# Distinct codes for environment structure and symmetry in postrhinal and retrosplenial cortices

Patrick A. LaChance [1] ✉ & Michael E. Hasselmo [1]

Complex sensory information arrives in the brain from an animal's first-person ('egocentric') perspective. However, animals can efficiently navigate as if referencing map-like ('allocentric') representations. The postrhinal (POR) and retrosplenial (RSC) cortices are thought to mediate between sensory input and internal maps, combining egocentric representations of physical cues with allocentric head direction (HD) information. Here we show that neurons in the POR and RSC of female Long-Evans rats are tuned to distinct but complementary aspects of local space. Egocentric bearing (EB) cells recorded in square and L-shaped environments reveal that RSC cells encode local geometric features, while POR cells encode a more global account of boundary geometry. Additionally, POR HD cells can incorporate egocentric information to fire in two opposite directions with two oppositely placed identical visual landmarks, while only a subset of RSC HD cells possess this property. Entorhinal grid and HD cells exhibit consistently allocentric spatial firing properties. These results reveal significant regional differences in the neural encoding of spatial reference frames.

Place-based learning and navigation require the translation of first-person ('egocentric') sensory information into a map-like ('allocentric') reference frame[1–6]. This process may require the brain to combine information about the egocentric bearings and distances of physical cues with information about an animal's allocentric head direction (HD; Fig. 1A, B)[2,7–10], resulting in an allocentric location signal similar to those exhibited by hippocampal place[11] and entorhinal grid[12] cells. Two brain regions that receive strong sensory input[13–16] and send extensive outputs to the hippocampal formation[17–20] are the postrhinal (POR) and retrosplenial (RSC) cortices. Both regions contain neurons that indicate an animal's spatial orientation relative to the surrounding environment in both egocentric[10,21,22] and allocentric[23,24] reference frames. However, despite anatomical differences in sensory and spatial inputs to each area[13,15,16,25], it is unknown whether POR and RSC neurons differ in their encoding of environmental geometry and landmark cues, and how these potential coding differences might impact downstream allocentric representations such as the entorhinal grid code. Answers to these fundamental questions are critical for

understanding how the brain distills a highly complex world into basis functions that can be manipulated to guide navigation and behavior.

In this work, we use tetrode recordings from single neurons in the POR and RSC of the same group of freely moving rats to demonstrate distinct neuronal responses in each brain region to the spatial cues that make up the local environment. Recordings in square and L-shaped environments reveal that egocentrically tuned POR neurons are largely sensitive to the global extent of boundary geometry, while RSC neurons respond to local features such as flat edges and corners. Further, recordings in square environments with one or two salient visual cues reveal that POR HD cells strongly incorporate egocentric visual information, firing in two opposite directions with two opposite cues, whereas RSC contains two distinct populations of HD cells, only one of which fires in two directions. In contrast, HD and grid cells recorded from the medial entorhinal cortex (MEC) or parasubiculum (PaS) maintain clear allocentric firing properties despite environmental manipulations. The significant differences among these brain regions provide insight into how different streams of egocentric information in

[1]Center for Systems Neuroscience and Department of Psychological and Brain Sciences, Boston University, Boston, MA, USA. ✉e-mail: plachanc@bu.edu

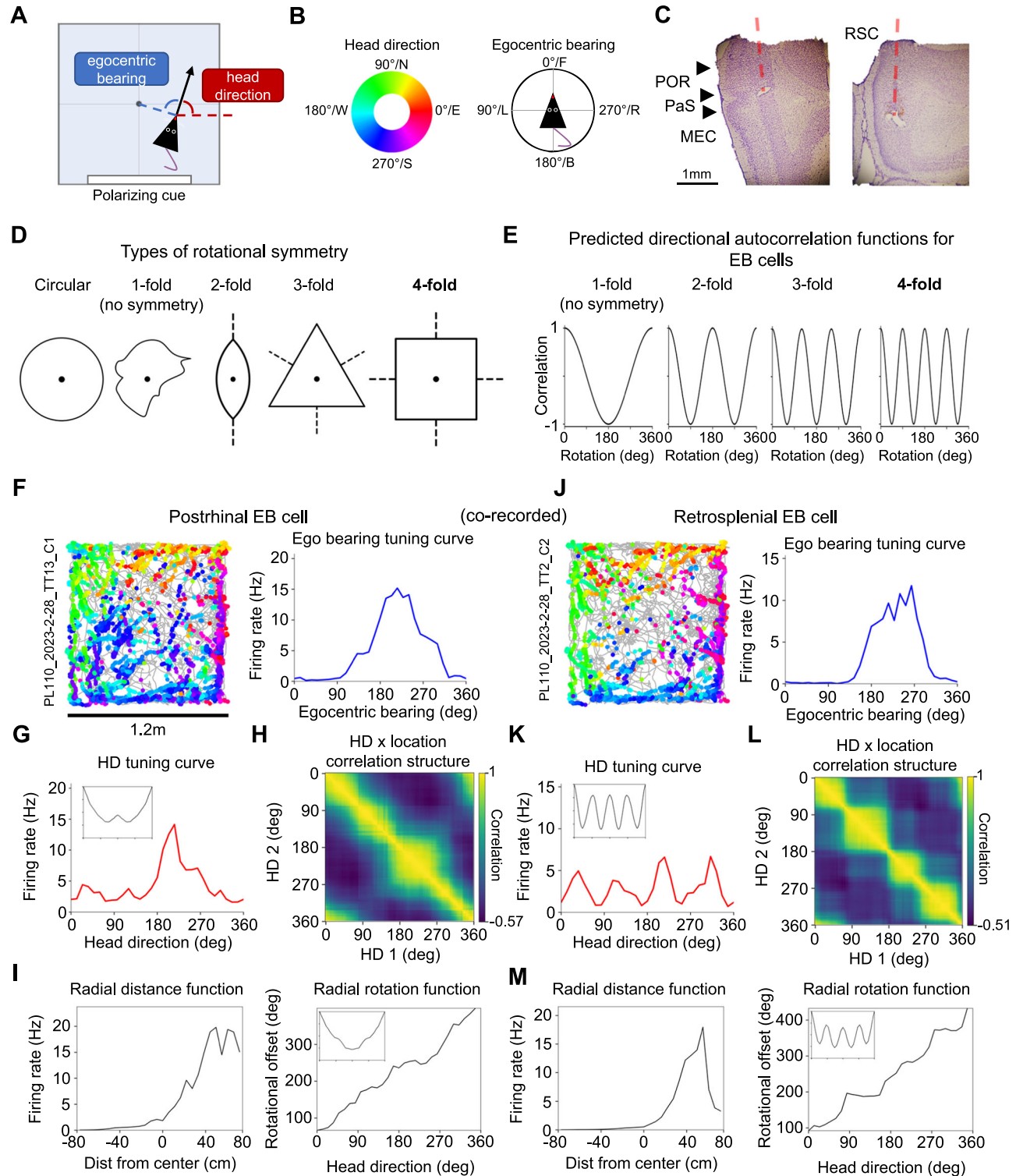

**Fig. 1 | Distinct egocentric responses to environmental symmetry in POR and RSC. A** Top-down schematic illustrating the measurement of allocentric head direction (relative to East) and egocentric bearing (relative to the environment center). **B** Color code for mapping of degrees to head direction (HD) (left) and compass for measuring egocentric bearing (right). **C** Nissl-stained sagittal section (*left*) and coronal section (*right*) from one rat showing electrode tracks (red dotted lines) through regions of interest. Histology from the five additional rats used in the study is included in Supplementary Fig. 15. **D** Example shapes with different degrees of rotational symmetry. Black dot represents the center-of-mass (centroid) of the shape. **E** Predicted directional autocorrelation functions for EB cells that may encode different degrees of rotational symmetry. Note that four-fold symmetry is

the most anticipated result for cells that encode local geometric features in a square environment. **F** Directional spike plot for an example POR EB cell (*left*) recorded in the square environment, along with its egocentric bearing tuning curve (*right*). (**G**) HD tuning curve for the cell in (**F**), with autocorrelation function inset. **H** HD x location correlation matrix for the cell in (**F**). **I** GLM-derived radial distance function (*left*) and rotation function (*right*) for the cell in (**F**), with autocorrelation function inset alongside the rotation function. **J–M** Same as (**F–I**) but for a simultaneously recorded RSC EB cell. Note the presence of strong four-fold symmetry across all domains for the RSC EB cell but not the POR EB cell (except for radial distance).

POR and RSC may interact with and support purely allocentric signals in connected regions such as MEC.

## Rotational symmetry coding in a square environment

We recorded 283 POR neurons and 519 RSC neurons from the same group of rats ($n = 6$; Fig. 1C) as they freely foraged for sugar pellets in a square ($120 \times 120$ cm) enclosure with a single polarizing cue (large white card) placed along the south wall (Fig. 1A). We used cross-validation with a generalized linear model (GLM)[10,26] as well as a shuffle distribution (see Methods) to classify 85 POR cells (30%) and 210 RSC cells (40%) as encoding the egocentric bearing (EB) of the center or boundaries of the environment (EB cells; Fig. 1A, B; Supplementary Fig. 1A,B). POR EB cells tended to have broad tuning profiles[10] (mean MVL: 0.32), whereas RSC EB cells were somewhat more sharply tuned (mean MVL: 0.40; Supplementary Fig. 1C). Many EB cells in both regions were classified as conjunctively encoding at least one other behavioral variable, including HD (POR: 60%, RSC: 33%), egocentric distance of the environment center/boundaries (POR: 22%, RSC: 21%), and linear speed (POR: 24%, RSC: 24%) or combinations of these variables as shown in Supplementary Fig. 1A,B. Strength of tuning to each of these variables varied along a continuum from non-tuned to strongly tuned, suggesting that significantly tuned cells may not constitute distinct 'cell types' despite showing consistent responses to a given variable (Supplementary Fig. 2).

Previous studies recording separately from these regions in square environments have suggested that POR EB cells are sensitive to the global geometry of the surrounding environment (equivalent to coding of the environment center)[10], while RSC EB cells are more sensitive to local geometric features such as the edges and corners of boundaries[21,27]. However, these two populations have not been directly compared in the same animals. We leveraged the inherent four-fold rotational symmetry of the local geometric features (i.e. walls and corners) in a square environment (Fig. 1D) to assess these sensitivities in each region, and contrast this with global coding of the environment's extended boundary geometry.

Four-fold symmetry of spatial firing was assessed across three domains (see Methods, Four-fold symmetry analyses): 1) four-fold symmetry in each cell's HD tuning curve (Fig. 1G, K; Fig. 2A, B; Supplementary Fig. 3), such that EB cells tuned to local geometric features of the square environment walls should fire preferentially in four discrete head directions; 2) four-fold symmetry in the correlation structure of location preferences across different head directions (computed by creating an array of firing rate maps for periods when the animal faced different HDs and calculating pairwise correlations between the rate maps; Fig. 1H, L; Fig. 2C, D; Supplementary Fig. 3), such that each of the four encoded head directions should be associated with a discrete firing location; and 3) four-fold radial symmetry about the center of the environment (Fig. 1I, M; Fig. 2E, F; Supplementary Fig. 4), such that the four discrete firing fields related to local features such as walls and corners should be located at 90° offsets with respect to the environment center. Symmetry was assessed in each domain by computing an autocorrelation function (Fig. 1E), which was used to compute a symmetry score (similar to a grid score[12]; see Methods). In addition to the expected four-fold symmetry, we also computed scores for one-fold, two-fold, and three-fold symmetry (Fig. 1E). While we expected RSC EB cells to display strong four-fold symmetry related to local geometric features, we expected POR EB cells to lack strong periodic symmetry, which may manifest in a larger population of cells showing one-fold symmetry (i.e., any deviation from circular symmetry would be non-periodic).

As expected, RSC EB cells tended to show strong four-fold symmetry across all domains (Fig. 1J–M, 2A–F; Supplementary Fig. 5), while POR EB cells did not (Fig. 1F–I, 2A–F; Supplementary Fig. 6). We computed an aggregate score, which combined the symmetry scores across the three domains, and considered EB cells with aggregate

scores that exceeded the 95th percentile of a shuffle distribution to be 'strongly symmetrical'. For RSC EB cells, 109/210 (52%) of RSC EB cells crossed this threshold for four-fold symmetry, while only 6/85 (7%) of POR EB cells did (Fig. 2G). In contrast, while only 8/210 (4%) of RSC EB cells showed significant one-fold symmetry, 22/85 (26%) of POR cells did (Fig. 2G). Neither region exhibited strong evidence for two-fold (POR: 8%, RSC: 7%) or three-fold (POR: 2%, RSC: <1%) symmetry. While 48% of RSC EB cells did not pass the threshold for strong four-fold symmetry, those non-symmetrical RSC cells still had significantly higher four-fold symmetry scores than the full POR EB cell population (Fig. 2H), suggesting that even RSC cells with subthreshold four-fold symmetry scores are distinct from POR cells in their encoding of environmental symmetry. Overall, RSC EB cells tend to exhibit strong four-fold radial symmetry in a square environment, indicative of coding for local geometric features such as walls and corners, while POR EB cells tend to lack periodic symmetry, indicative of coding for global properties of environmental geometry.

## Local vs. global coding in an L-shaped environment

To further put local and global geometric cues in conflict, we recorded EB cells (46 POR, 177 RSC) as animals ($n = 6$) foraged in both the $120 \times 120$ cm square environment, and the same environment with additional walls inserted to transform it into an L-shape (Fig. 3A). To assess potential novelty effects of the inserted walls, two of the six animals had been extensively habituated to the L-shape but not the square (so they received the following set of sessions: L-shape 1, Square, L-shape 2), while the other four had been extensively habituated to the square but not the L-shape (Square 1, L-shape, Square 2).

We used a previously developed GLM framework[28] to distinguish between local and global egocentric tuning in both the square and L-shape (Fig. 3B), with a globality index (GI) indicating whether EB cells were more strongly tuned to the environment centroid (global tuning; GI > 0) or nearby walls (local tuning; GI < 0). As these representations are quite similar in a square environment[21,28,29], we looked specifically at the change in GI from the square to the L-shape. In agreement with the symmetry analyses, POR EB cells tended to shift toward higher GI values (more global; Fig. 3C, D; Supplementary Fig. 8), while RSC EB cells shifted toward lower GI values (more local; Supplementary Fig. 7). The same pattern was observed for animals extensively habituated to either the square or L-shape (Supplementary Fig. 9), suggesting that novelty did not strongly impact the egocentric tuning preferences in either brain region.

In some animals ($n = 3$; all habituated to square), tetrodes were advanced beyond the ventral POR border and grid cells ($n = 64$) were recorded from the adjacent MEC or PaS in both the square and L-shaped environments (Fig. 3E; Supplementary Fig. 10). To compare EB cell responses to the L-shape with those of grid cells in these downstream areas, we assessed the change in firing rate for each group of cells in the vicinity of the inserted walls (within 20 cm). Both POR and RSC EB cells showed a statistically significant increase in firing rate near the walls (but not away from the walls), while grid cells showed no overall change in firing rate (Fig. 3F, G), suggesting distinct mechanisms underlying EB cell and grid cell encoding of environmental geometry. Interestingly, while the RSC increase in firing rate was observed regardless of extensive habituation to either the square or L-shape, only the POR EB cells of animals initially habituated to the square (but not those habituated to the L-shape) showed a significant change (Fig. 3H, I), suggesting that the increased firing rate of POR EB cells near the inserted walls may be at least partially a result of novelty.

## Incorporation of visual landmarks into the HD signal

The formation of allocentric spatial representations likely requires the integration of egocentric representations with an allocentric HD signal. Cells with HD correlates have been reported in all brain regions investigated in the current study[23,24,30]. Of cells recorded in the square

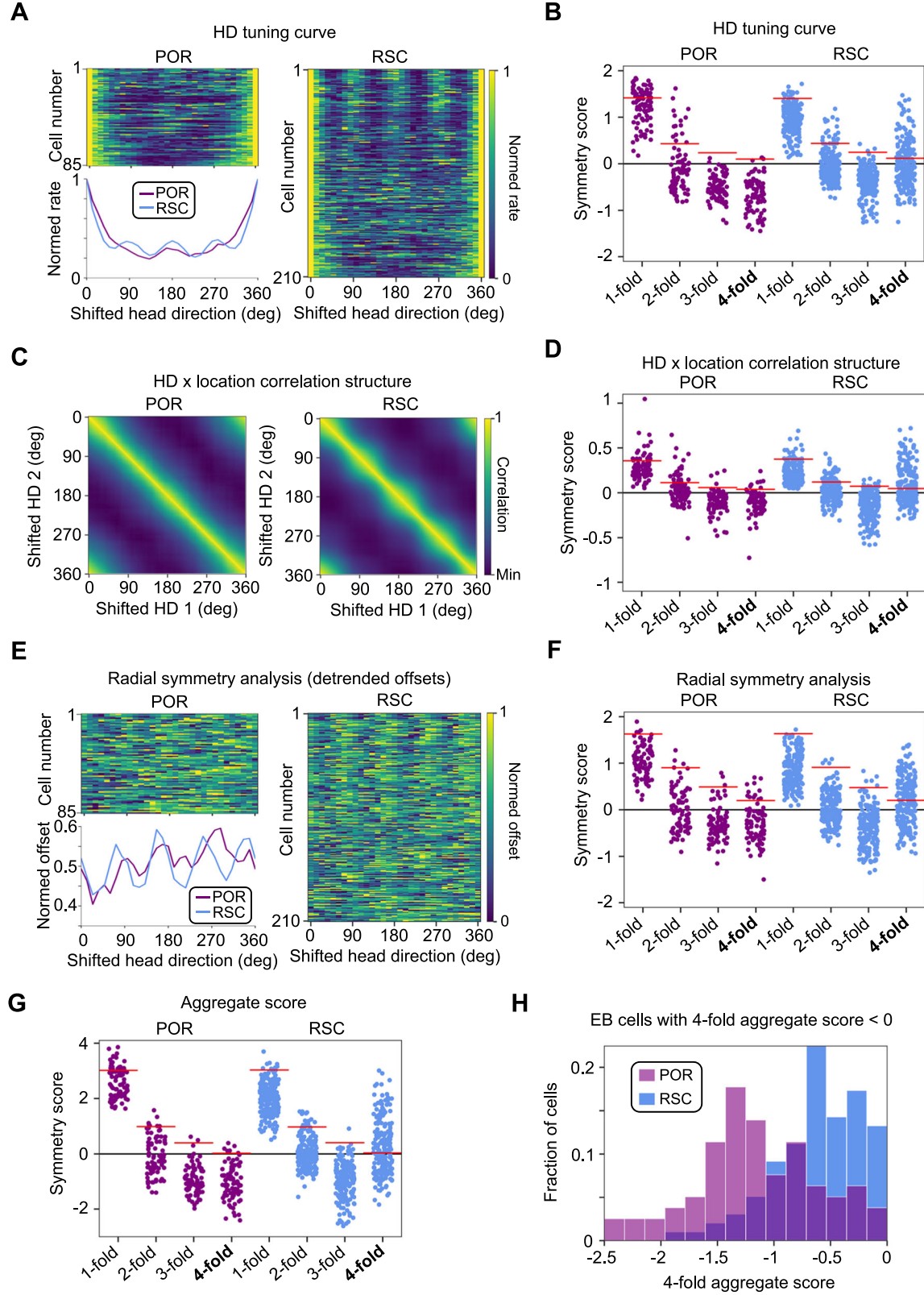

environment, HD cells included 97/283 POR cells (34%), 122/512 RSC cells (24%), and 65/223 MEC cells (29%), meaning that they tended to fire most when the animal faced one particular HD (preferred firing direction; PFD; Fig. 1A; Fig. 4A; Supplementary Fig. 1). POR HD cells tended to have broad tuning profiles (mean MVL: 0.32), while RSC and MEC/PaS HD cells resembled more 'classic' HD units with narrow tuning curves[31] (RSC mean MVL: 0.49; MEC mean MVL: 0.59;

Supplementary Fig. 2, 11). POR HD cells could be further separated into two categories, depending on their PFDs. Those with PFDs oriented toward the familiar visual landmark tended to have a relatively sharp peak in their tuning curves, while those with PFDs oriented away from the landmark had relatively sharp troughs (distinguished by comparing the $R^2$ of upright vs. inverted von Mises curves fit to the tuning curves[23]; see Methods; Supplementary Fig. 11D, E). This result suggests

**Fig. 2 | Population coding of rotational symmetry. A** Normalized HD tuning curves for all POR egocentric bearing (EB) cells (*top left*) and RSC EB cells (*right*) recorded in the square environment, shifted for each cell such that the maximum firing rate lies at 0° ('shifted head direction'). *Bottom left*, mean tuning curve for each brain area. **B** One-fold, two-fold, three-fold, and four-fold symmetry scores for the HD tuning curves of all RSC ($N = 210$) and POR EB ($N = 85$) cells. Red lines indicate the 95th percentile of a shuffle distribution. **C** Mean HD x location correlation matrix for all POR EB cells (*left*; $N = 85$) and RSC EB cells (*right*; $N = 210$) recorded in the square environment. **D** Same as (**B**) but based on HD x location correlation matrices ($N = 85$ POR cells, 210 RSC cells). **E** Normalized detrended GLM-derived rotation functions for all POR EB cells (*top left*; $N = 85$) and RSC EB cells (*right*; $N = 210$) recorded in the square environment. *Bottom left*, mean detrended rotation function for each brain area. **F** Same as (**B**) but based on GLM-derived rotation functions ($N = 85$ POR cells, 210 RSC cells). **G** Aggregate symmetry scores based on summing individual scores from (**B, D, F**) ($N = 85$ POR cells, 210 RSC cells). **H** Histogram of four-fold symmetry scores for RSC and POR EB cells that did not pass threshold for significant four-fold symmetry ($N = 79$ POR cells, 101 RSC cells). Note that the RSC population is still significantly rightward-shifted compared to the POR population (dark purple color shows overlap between the distributions; two-sided Wilcoxon rank-sum test, $Z = 5.41$, $P = 6.14\text{e-}8$). Source data are provided as a Source Data file.

that POR HD cells strongly incorporate egocentric information regarding visual landmarks. In contrast, this distinction was not present for RSC or MEC/PaS HD cells, which always exhibited strong peaks regardless of whether the PFD was oriented toward or away from the landmark (Supplementary Fig. 11E).

To further probe the incorporation of egocentric landmark signals into the HD correlates of each region, for the four animals habituated to the square environment, we duplicated a familiar landmark (cue A, a white cue card along the south wall) along the opposite wall (cue B; session order: A1, AB, A2; Fig. 4B; Supplementary Fig. 12). In this AB condition with two cues, POR HD cells ($n = 34$) became strongly bidirectionally tuned (Fig. 4C, D; Fig. 5A), although their firing rate modulation by cue B was consistently smaller in magnitude than their modulation by cue A (Fig. 4E). Interestingly, POR HD cells remained slightly bidirectional in the A2 session, though to a lesser extent than the AB session (Fig. 4D). Like POR, the overall population of RSC HD cells ($n = 37$) became significantly bidirectional in the AB session, and displayed a similar effect to the POR cells wherein they remained slightly bidirectional in the A2 session, though to a lesser extent than the AB session (Fig. 4F, G; Fig. 5B). As with POR, RSC HD cells were more strongly modulated by the more familiar cue A than cue B in the AB session (Fig. 4H). However, unlike in POR and RSC, HD cells in MEC/PaS ($n = 30$) did not become bidirectional in the AB session (Fig. 4I, J; Fig. 5C).

Despite the population shift toward bidirectionality, RSC HD cells appeared to be segregated into a unidirectional subpopulation and a bidirectional subpopulation (Fig. 4G; Fig. 5B). We found we could predict which RSC cells would become bidirectionally tuned according to the width of their extracellular waveforms, such that cells with narrow waveforms (<200-μs peak-trough latency) did not become bidirectional, while those with broader waveforms became bidirectional to varying degrees (similar to previous findings in multicompartment environments[32]; Fig. 5D). Looking at the full population of HD cells recorded in the square session, we found that cells with narrow waveforms tended to have higher MVLs and peak firing rates in RSC (MVLs: narrow mean = 0.75, wide mean = 0.39; peak firing rates: narrow mean = 17.41 Hz, wide mean = 6.26 Hz) and MEC/PaS (MVLs: narrow mean = 0.76, wide mean = 0.45; peak firing rates: narrow mean = 6.99 Hz, wide mean = 3.55 Hz), but not POR (Supplementary Fig. 13). This result suggests that cells with narrow waveforms may be morphologically distinct and fire more similarly to 'classic' HD cells[31].

In addition to HD cells, we also recorded 61 MEC/PaS grid cells in the visual landmark experiment, which did not change their firing patterns (Supplementary Fig. 14), again suggesting distinct mechanisms underlying POR/RSC HD signals and downstream MEC/PaS grid cell representations.

## Discussion

The results of this study indicate distinct but complementary egocentric representations of environmental structure and symmetry in POR and RSC, which are further distinct from the largely allocentric firing properties of downstream MEC/PaS cells. RSC EB cells are strongly influenced by local geometric features, while POR EB

cells largely appear to reference their firing to the global structure of the environment (Fig. 6A–F). The more global sense of space in POR may be particularly suited to drive global allocentric spatial firing in the downstream MEC/PaS, which was largely unaffected by changes in local geometric features in our data (though grid cells can be impacted by environmental geometry in other ways[33,34]). However, in contrast with HD and grid cells in MEC/PaS, POR HD cells were strongly modulated by duplication of a visual landmark (becoming bidirectional), while RSC HD cells appeared to fall into two discrete populations that either did or did not respond to the duplicated landmark, whereas MEC HD cells did not show bidirectional firing (Fig. 6G–I) suggesting a complex transformation between upstream egocentric and downstream allocentric spatial representations.

POR has been previously described as representing the surrounding environment in terms of global shape parameters, such as the centroid and principal axis (or slope) of the environment[2,10]. In this framework, the centroid is represented by POR EB cells (sometimes called 'center-bearing cells') and involves computing a vector average of the distances and bearings of all physical cues in the environment[2,10], while the principal axis is represented by HD cells and appears to give an estimate of the animal's HD based on the global constellation of stable landmarks[2,10,35]. Duplicating a familiar landmark along the opposite wall introduces uncertainty into that estimate, causing the HD cells to fire in two opposite directions[23]. However, while the current study reinforces the notion that that POR EB cells are sensitive to a more global account of environmental geometry than RSC EB cells, future experiments in more complex environments (e.g., trapezoid or asymmetric triangle) will be necessary to determine if POR EB cells are truly computing the environment centroid or if they more closely follow some other global boundary-encoding framework. For example, in a V-shaped or similar environment where the centroid lies outside the navigable space, POR EB cells may encode the medial axis of the shape[36] instead of representing a single centroid, or could potentially display more local geometric coding than that observed in square and L-shaped environments. It has been previously demonstrated that POR EB cells can vary in their tuning to local vs. global aspects of environmental geometry[28], so different cells may exhibit distinct responses.

In contrast to the more global egocentric code in POR, RSC has been described as representing specific local geometric features (such as walls and corners[21,27]) and specific visual landmarks[32,37,38], in addition to displaying periodic activation patterns in both open field[21] and track[39] environments. In agreement with this framework, RSC EB cells are highly sensitive to the inherent rotational symmetry of the local features of a square environment, and one population of RSC HD cells is sensitive to the duplication of a familiar visual landmark. The differences between global shape and local feature representations in POR and RSC, respectively, mirror decades of behavioral studies that provide evidence for both global and local accounts of spatial processing[36,40,41]. The current study indicates that these local and global reference frames are both present in the rodent brain, and that they are represented to different degrees in separate brain regions.

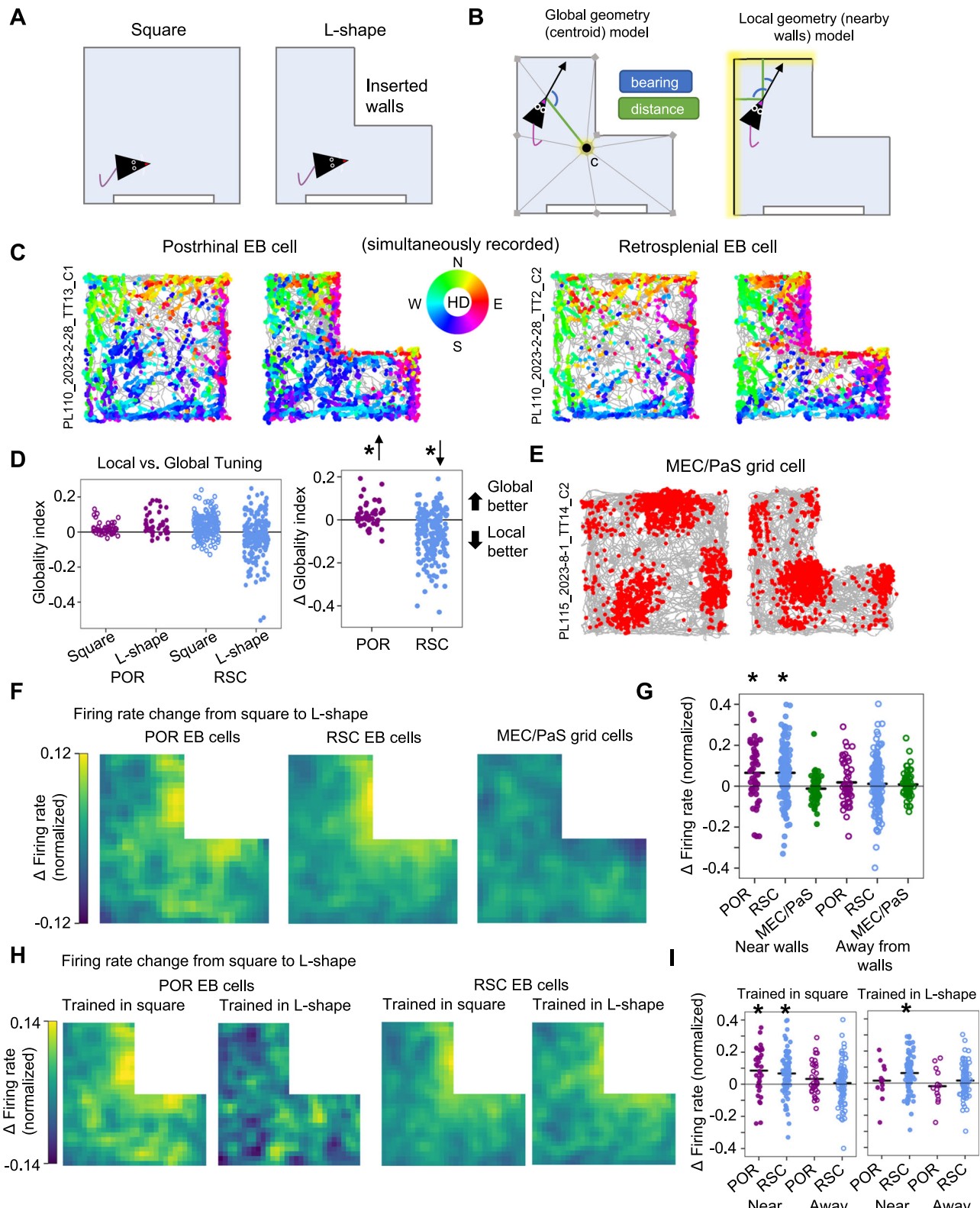

**A** Square | L-shape (Inserted walls)

**B** Global geometry (centroid) model | Local geometry (nearby walls) model — bearing / distance

**C** Postrhinal EB cell | (simultaneously recorded) | Retrosplenial EB cell

**D** Local vs. Global Tuning — Globality index / Δ Globality index — Square, L-shape (POR), Square, L-shape (RSC) — Global better / Local better — POR, RSC

**E** MEC/PaS grid cell

**F** Firing rate change from square to L-shape — POR EB cells | RSC EB cells | MEC/PaS grid cells — Δ Firing rate (normalized)

**G** Δ Firing rate (normalized) — POR, RSC, MEC/PaS (Near walls); POR, RSC, MEC/PaS (Away from walls)

**H** Firing rate change from square to L-shape — POR EB cells: Trained in square, Trained in L-shape | RSC EB cells: Trained in square, Trained in L-shape — Δ Firing rate (normalized)

**I** Δ Firing rate (normalized) — Trained in square: POR, RSC (Near), POR, RSC (Away); Trained in L-shape: POR, RSC (Near), POR, RSC (Away)

Differences between RSC and POR cells may result from their disparate inputs. Visual inputs to POR largely originate from superior colliculus (SC; via the lateral posterior thalamic nucleus (LP))[13,16], which is an evolutionarily ancient visual structure that has been linked to a number of ethologically relevant behaviors[42,43]. Recently, the rodent SC has been suggested to process optic flow stimuli associated with forward self-motion[44]. Optic flow along the boundaries of the environment may be integrated over time to produce an egocentric bearing signal in POR that can drive the animal to the center of the environment, similar to the honeybee's centering response[45–47]. Activity in brain regions that process optic flow stimuli has been shown to correlate with navigational performance in humans[48]. POR also receives afferentation from the HD cell-rich anterior thalamus (ATN)[49,50], although this input is minor compared to LP[13]. In contrast, RSC appears to receive strong convergent inputs from primary visual cortex (V1) and the ATN[15,25,51], which may explain both the 'classic' HD tuning[31] in

**Fig. 3 | Distinct encoding of global vs. local geometry by POR and RSC EB cells.**
**A** Top-down schematic showing the transformation of the square into an L-shape by inserting walls. **B** Top-down schematic of the frameworks used to model cell firing based on a vector average of all geometric features (global/centroid model, *left*) or only features within a certain distance of the animal (local/nearby walls model, *right*). **C** Directional spike plots (with color code for head direction (HD) on right) for a simultaneously recorded POR egocentric bearing (EB) cell and RSC EB cell pair in both square and L-shape sessions (same cells from Fig. 1F−M). **D** Plot of globality indices (GI) for all POR ($N = 46$ cells) and RSC ($N = 177$) EB cells recorded in square and L-shaped environments (*left*), along with the change in GI between the square and L-shape (*right*). Note that POR EB cells tend toward global tuning, while RSC EB cells tend toward local tuning (two-sided Wilcoxon signed-rank tests, POR: $W = 289$, $P = 5.33e-3$; RSC: $W = 2986$, $P = 7.84e-13$). * indicates $P < 0.05$. **E** Path and spike plot for an MEC/PaS grid cell recorded in square and L-shaped environments. **F** Allocentric location firing rate maps showing the average change in normalized firing rates between the square and L-shape for POR EB cells (*left*), RSC EB cells (*middle*), and MEC/PaS grid cells (*right*). **G** Change in firing rate for individual cells (POR EB cells ($N = 46$), RSC EB cells ($N = 177$), MEC/PaS grid cells ($N = 64$)) between the square and L-shape near (<20 cm) or far from (> 20 cm) the inserted walls. Note that POR (two-sided Wilcoxon signed-rank test, $W = 273$, $P = 2.92e-3$) and RSC ($W = 3299$, $P = 2.01e-11$) EB cells, but not MEC/PaS grid cells (all remaining comparisons $P > 0.05$), show increased firing near the inserted walls. * indicates $P < 0.05$. **H** Allocentric location rate maps showing the change in normalized firing rate for POR EB cells (*left*) and RSC EB cells (*right*) between the square and L-shape, separated into animals initially trained in either the square or L-shape. **I** Change in normalized firing rate near or far from the inserted walls for POR and RSC EB cells of animals trained in either the square (*left*; $N = 33$ POR cells, two-sided Wilcoxon signed-rank test, near walls: $W = 118$, $P = 2.92e-3$; away from walls: $P > 0.05$; N = 89 RSC cells, near walls: $W = 856$, $P = 2.72e-6$; away from walls: $P > 0.05$) or L-shape (*right*; $N = 13$ POR cells, two-sided Wilcoxon signed-rank test, near walls: $P > 0.05$; away from walls: $P > 0.05$; $N = 88$ RSC cells, near walls: $W = 804$, $P = 1.57e-6$; away from walls: $P > 0.05$). Note that POR EB cells do not show increased firing near the inserted walls for animals trained in the L-shape. * indicates $P < 0.05$. Source data are provided as a Source Data file.

RSC as well as the attraction of RSC EB cells to specific local geometric features such as the edges and corners of boundaries, which may be conveyed by edge detectors in V1[52]. RSC egocentric boundary cells have been modeled in this way previously[53].

POR and RSC both send projections to the entorhinal cortex and hippocampus[17–20]. POR (as with its human homolog, the parahippocampal cortex; PHC) is known to send particularly dense inputs to MEC[54,55]. Grid cells in MEC are thought to embody a global allocentric distance code[56] that is mostly unaffected by local geometric features, and this code may be best supported by inputs from POR that reflect the animal's position with respect to the global structure of the environment. Likewise, the main hippocampal structure targeted by POR is the subiculum[20], whose cells are known to represent allocentric space with similar firing patterns in environments with different local geometric structures[57]. As a complement to these global POR inputs, the differential RSC representation of local geometric features may be useful for anchoring downstream allocentric spatial representations to particular physical cues. This principle has been previously used to model the formation of grid cell firing patterns based on inputs from RSC EB cells that code for local environmental features[58]. Further, the communication between POR and RSC may create an efficient representation of environmental geometry that relates specific local geometric features to each other by computing their position with respect to the environment centroid[59].

In humans, damage to both PHC and RSC results in topographical disorientation[60–62], which could result from disrupting the associations between egocentric and allocentric directional processing explored in the current study. In particular, RSC damage has been associated with deficits in orienting with respect to specific known landmarks[62], while PHC damage has largely resulted in patients being unable to orient themselves in new environments[61]. These findings align with our current results that RSC cells are sensitive to specific geometric features and landmarks that may be helpful for orienting in familiar environments, while POR cells compute a more general sense of surrounding space that may be more useful for orienting in new environments where no familiar landmarks have been established.

## Methods
### Subjects
Subjects were 6 female Long-Evans rats (Charles River Laboratories) aged 4-7 months and weighing 265–335 g prior to surgery. Rats were individually housed in Plexiglas cages and maintained on a 12 h light/dark cycle. Prior to surgery, food and water were provided ad libitum. All experimental procedures involving the rats were performed in compliance with institutional standards as set forth by the National Institutes of Health *Guide for the Care and Use of Laboratory Animals*

and approved by the Boston University Institutional Animal Care and Use Committee.

### Electrode construction
Each animal was implanted with two moveable microdrives that each consisted of a bundle of eight tetrodes. Tetrodes were constructed by twisting together four strands of 17-μm nichrome wire (Alleima), which were subsequently threaded through a single 26-gauge piece of polyimide tubing affixed to the shuttle of a 3D-printed microdrive[63] (print designs taken from https://github.com/buzsakilab/3d_print_designs) which could be advanced in the dorsal-ventral plane by turning a single 00-90 screw. The end of each wire was connected to one contact of a 32-channel electrode interface board (EIB; Neuralynx).

### Electrode implantation
Animals were anesthetized with isoflurane. They were then placed into a stereotaxic frame, and an incision was made in the scalp to expose the skull. Two craniotomies were drilled: one above the retrosplenial cortex (6 mm posterior and 0.5 mm lateral to bregma), and another above the postrhinal cortex (1.9 mm posterior and 4.6 mm lateral to lambda). Anchor screws were also drilled into the skull and secured with a layer of metabond. One anchor screw placed above the cerebellum or frontal cortex was used as a ground screw. The retrosplenial-targeting tetrode bundle was implanted 6 mm posterior to bregma and immediately lateral to the midline sinus, and just ventral enough for the tetrode tips to penetrate the cortical surface (-0.5 mm). This bundle was also given a 10° angle in the medial-lateral plane, such that the tetrode tips pointed medially. The postrhinal-targeting bundle was implanted 4.6 mm lateral to lambda, 0.40 mm anterior of the transverse sinus, and -0.5 mm ventral of the cortical surface. This bundle was given a 10° angle in the anterior-posterior plane, such that tetrode tips were pointed forward. Both drive bodies were secured to the skull using dental acrylic, and were surrounded by a 3D-printed headcap[63] which housed the EIBs and was also secured to the skull using dental acrylic.

### Recovery and behavioral training
Rats were allowed 7 days to recover from surgery, after which they were placed on food restriction such that their body weight reached 85–90% of their pre-surgical weight. During this time, the rats were also trained to forage for randomly scattered sucrose pellets (20 mg, chocolate flavor) within a 120 × 120 cm box with a black floor and 60 cm high black walls that was surrounded on all sides by a floor-to-ceiling circular black curtain. For two of the rats, additional walls were placed into the environment to block the northeast quadrant, such that those rats were habituated to an L-shaped environment instead of the square environment. The box itself was featureless except for a

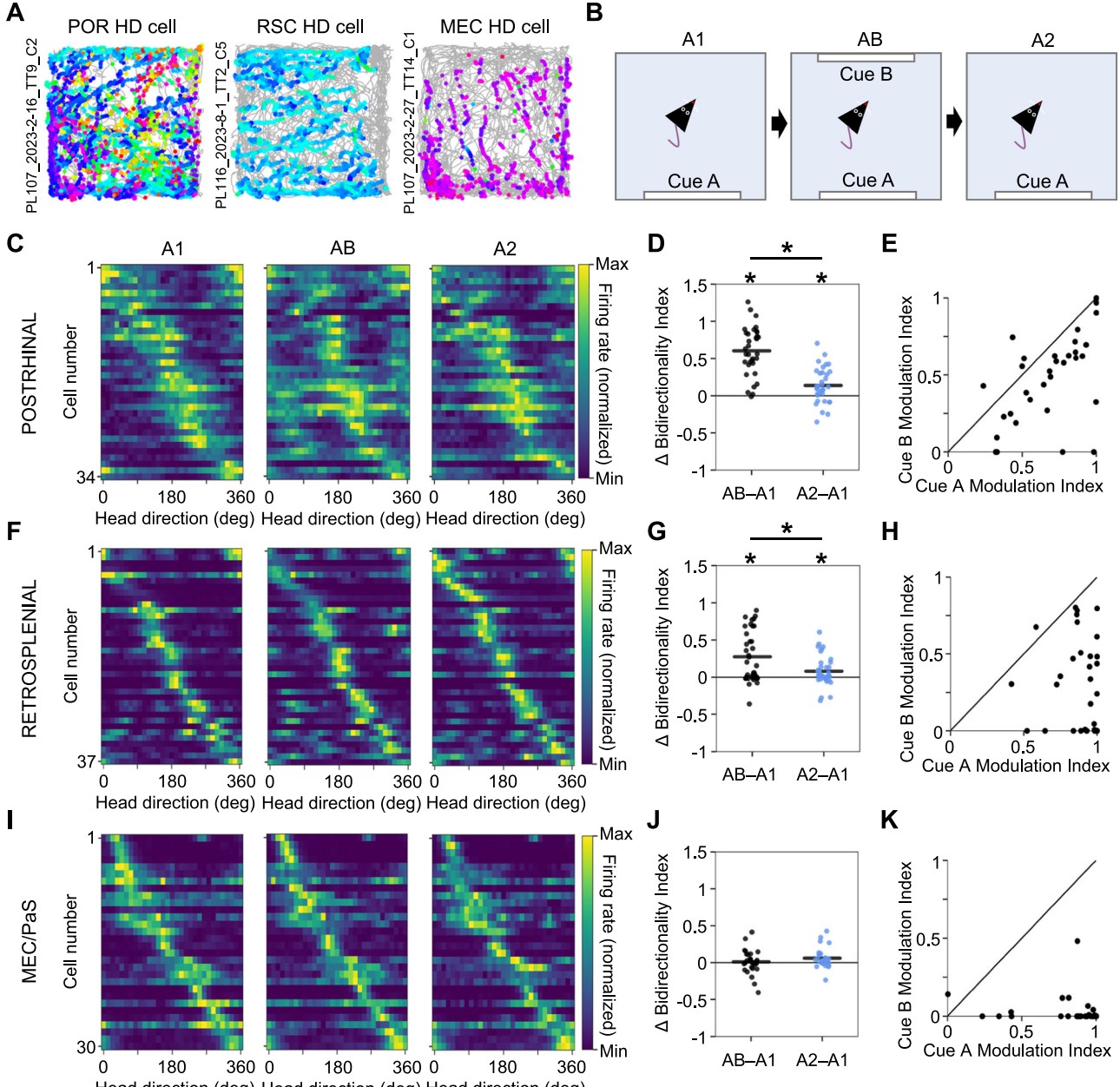

**Fig. 4 | Distinct HD cell responses to visual landmarks. A** Directional spike plots for example head direction (HD) cells recorded in the square environment from POR (*left*; N = 34 cells), RSC (*middle*; N = 37 cells), and MEC/PaS (*right*; N = 30 cells). **B** Top-down schematic illustrating the three sessions of the AB experiment. **C** Normalized HD tuning curves for POR HD cells across all three sessions of the AB experiment. **D** Change in bidirectionality for POR HD cells between the A1 session and both the AB and A2 sessions (two-sided Wilcoxon signed-rank test, AB vs. A1: W = 1, P = 2.33e-10; A1 vs. A2: P = 118, P = 1.57e-3; AB vs. A2, W = 9, P = 3.84e-9). *

indicates P < 0.05. **E** Comparison of the degree of firing rate modulation attributed to cue A or cue B for all POR HD cells recorded in the two-cue experiment (two-sided Wilcoxon signed-rank test, W = 49, P = 3.53e-5). **F–H** Same as (**C–E**) but for RSC HD cells (two-sided Wilcoxon signed-rank tests, bidirectionality: A1 vs. AB, W = 111, P = 1.41e-4; A1 vs. A2, W = 187, P = 0.012; AB vs. A2, W = 171, P = 5.62e-3; modulation index: W = 3, P = 7.28e-11). **I–K** Same as (**C–E**) but for MEC/PaS HD cells (two-sided Wilcoxon signed-rank test, W = 207, P = 0.62). Source data are provided as a Source Data file.

single white cardboard sheet (cue A) placed along the south wall. The cue card was 50 cm in height and had a width of 72 cm, such that it covered 60% of center of the horizontal extent of the wall. Neural recordings began once the animals started showing uniform coverage (>80%) of the entire arena.

**Baseline recording sessions**

Over weeks or months, tetrodes were 'screened' for units that displayed well-isolated waveforms as the animals foraged for sugar pellets in the arena. Electrical signals were pre-amplified using unit-gain operational amplifiers on an HS-36-LED headstage and sent to a Digital Lynx SX acquisition system (Neuralynx). Signals from each tetrode wire were then differentially referenced to a relatively quiet, low-noise channel from a separate tetrode and bandpass filtered (600 Hz to 6 kHz) using Cheetah acquisition software (Neuralynx). If signals on a given tetrode exceeded a pre-defined amplitude threshold (typically 30 to 50 µV), they were time-stamped and digitized at 32 kHz for 1 ms. The headstage was also equipped with red and green light-emitting diodes (LEDs) spaced approximately 6 cm apart over the head and back of the animal, respectively. A color video camera positioned over the arena captured video frames with a sampling rate of 30 Hz, and these were timestamped so they could be matched to the neural data.

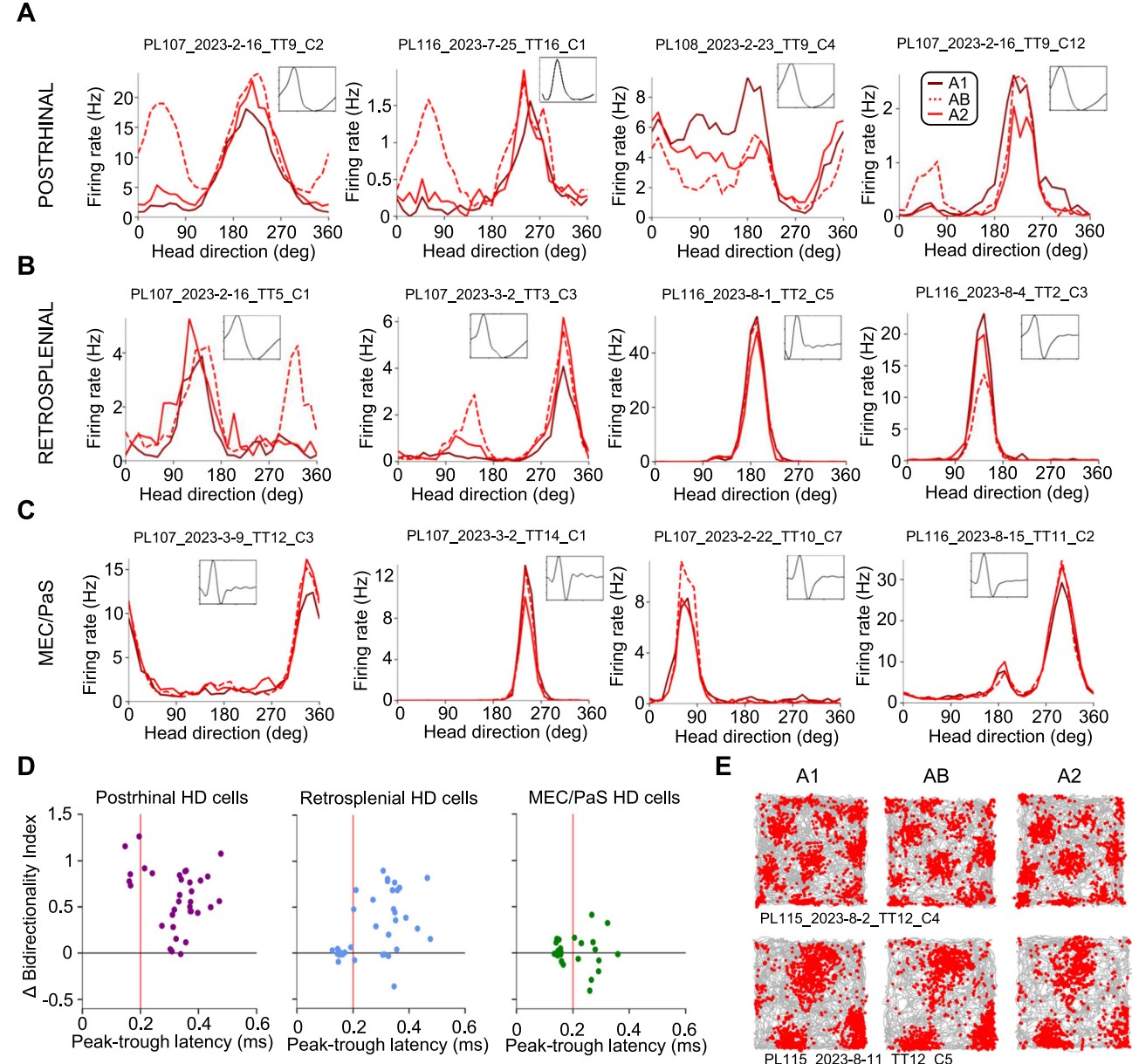

**Fig. 5 | Extracellular waveforms predict HD cell bidirectional symmetry in RSC. A** HD tuning curves for four example POR head direction (HD) cells recorded across all three sessions of the AB experiment. Average extracellular waveform for each cell is inset into each plot. **B** Same as (**A**) but for RSC HD cells. **C** Same as (**A**) but for MEC/PaS HD cells. **D** Comparison of extracellular waveform width vs. change in bidirectionality in the AB session for POR (*left*; *N* = 34), RSC (*middle*; *N* = 37), and MEC/PaS (*right*; *N* = 30) HD cells. **E** Path and spike plots for two example grid cells recorded across all three sessions of the AB experiment, which did not change their firing properties.

Videos were further analyzed using DeepLabCut[64] to obtain an accurate estimate of the LED positions. If well-isolated waveforms were visually apparent, a 20 min baseline recording in the 120 cm square box (for 4 animals) or a 15 min baseline recording in the L-shaped environment (for 2 animals) took place. The difference in baseline time corresponded to the difference in area of the environment, so that time per unit area was constant. Otherwise, electrodes were advanced ~50 to 100 μm and screened again at least 2 hours later or the next day.

**Spike sorting**
Spike sorting was conducted offline, and began by automatically clustering the collected spikes for a recording session using the automated clustering program Kilosort[65]. If cells were recorded across multiple session in a day, automatic sorting was performed on a merged dataset to ensure cluster continuity, after which the sessions were separated again for manual curation and analysis. The manual step involved visualizing waveform features such as peak, valley, height, width, and principal components to visualize the characteristics of individual spikes across multiple tetrode wire as a 3D scatter plot (SpikeSort3D, Neuralynx). While not always required, adjustment of automatically sorted clusters was performed by either merging clusters or drawing a polygon around the visually apparent boundaries of each cluster. Single-unit isolation was assessed using metrics including L-ratio and isolation distance, as well as assessing temporal autocorrelograms for the presence of a refractory period. As long as tetrodes were advanced between recording sessions, cells recorded across days were treated as independent units. Otherwise, the recording session with the larger number of clusters was used for baseline analyses.

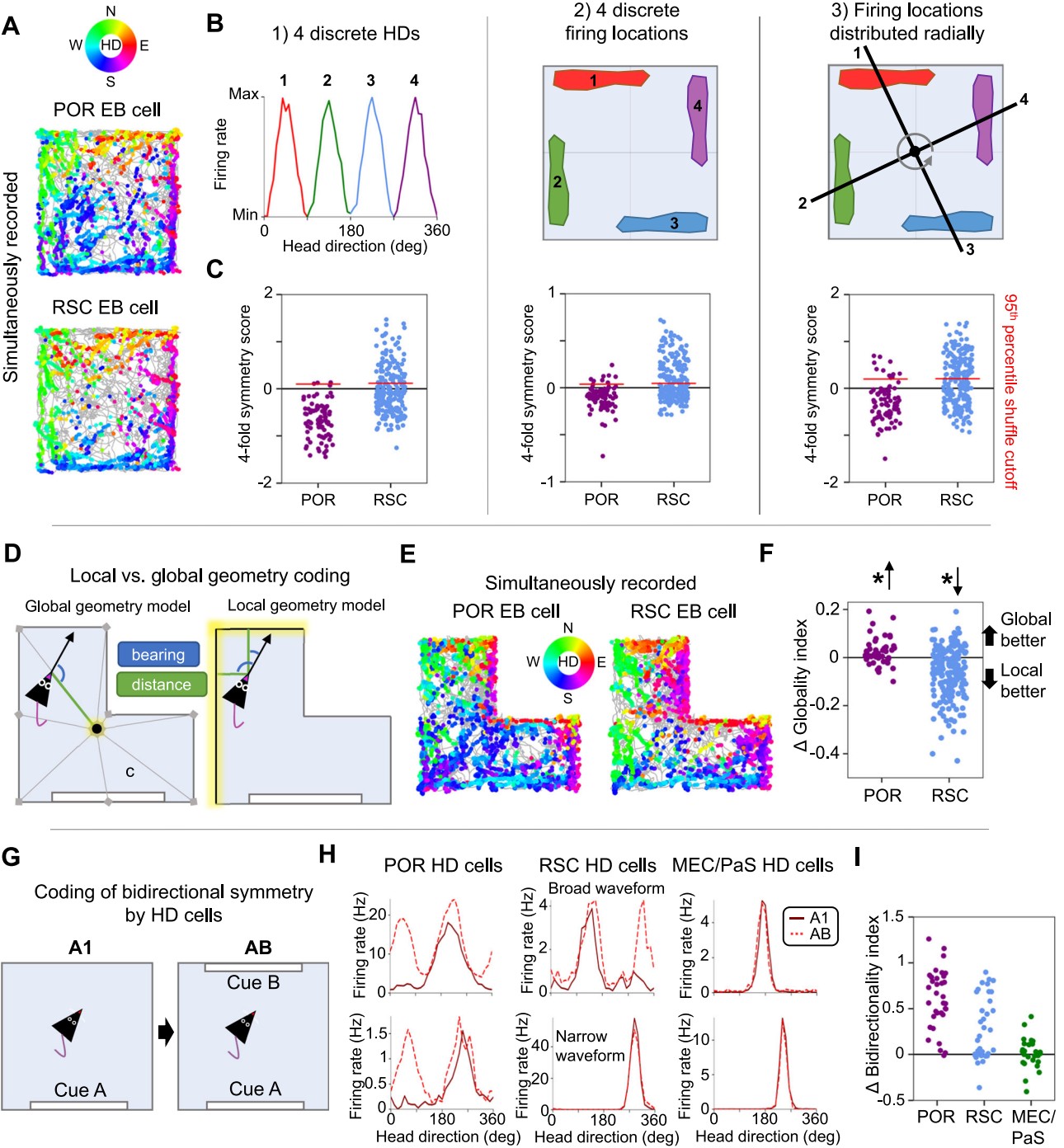

**Fig. 6 | Summary of results. A** Directional spike plots for two EB cells simultaneously recorded from POR and RSC. **B** Schematic illustrations of the three domains by which we assessed rotational symmetry of EB cell firing, specifically as it relates to four-fold symmetry. **C** Four-fold symmetry scores for the full POR and RSC EB cell population based on the three symmetry domains illustrated in (**B**) ($N = 210$ RSC cells, 85 POR cells). **D** Schematic illustration of models used to assess global vs. local geometric processing in an L-shaped environment. c = environment centroid. **E** Directional spike plots for the two cells from (**A**) recorded in an L-shaped environment. **F** Change in globality index between the square and L-shaped environments for the full POR ($N = 46$) and RSC ($N = 177$) EB cell population, demonstrating that POR cells shift toward global coding while RSC cells shift toward local coding. * indicates $P < 0.05$ for a two-sided Wilcoxon signed-rank test. **G** Schematic illustration of the cue duplication paradigm used to test coding of bidirectional symmetry by HD cells. **H** Example tuning curves for POR, RSC, and MEC/PaS HD cells recorded in the cue duplication experiment. **I** Degree of bidirectionality exhibited by POR, RSC, and MEC/PaS cells in the AB condition compared to the A1 condition.

## Histology
Once recordings were complete, animals were deeply anesthetized with sodium pentobarbital, and small marking lesions were made at the tetrode tips by passing a small anodal current ($20\,\mu A$, 7–10 s) through one active wire from each tetrode. Animals were then intracardially perfused with saline followed by 10% formalin solution, after which the brains were removed from the skull and postfixed in 10% formalin for at least 24 h. The brains were then transferred to 30% sucrose solution for at least 24 h, after which they were blocked such that the posterior portion of the brain could be sliced sagittally (for

POR and MEC/PaS placement) and the anterior portion could be sliced coronally (for RSC placement). The brains were frozen and sliced into 40-μm sections using a cryostat, with sections mounted onto glass microscope slides and subsequently stained with cresyl violet. Locations of individual cells were determined by measuring backward from the most ventral portion of the marking lesions or electrode tracks (if lesions were not visible; Supplementary Fig. 15). Delineations of parahippocampal regions were drawn mainly from[66,67].

### Environmental geometry manipulation recording sessions

On some days, particularly if EB cells or grid cells were suspected in the initial baseline recording session, cells were recorded during a subsequent session where the boundary geometry was changed. For animals trained in the square (4 rats) this involved placing additional walls into the environment to transform it into an L-shape for a 15 min recording session. This L-shape session was followed by a final session in the square environment for 20 min (session order: Square 1, L-shape, Square 2). For animals trained in the L-shape (2 rats), the second session was one in which the inserted walls were removed, transforming it into a square arena for a 20 min recording session. This square session was followed by reinserting the walls for a final 15 min L-shape session (session order: L-shape 1, Square, L-shape 2). Animals were removed from the recording environment between sessions and placed either into their home cage or a cardboard box outside the curtains while the environment was manipulated by the experimenter. The floor of the enclosure was also wiped down between sessions using veterinary-safe disinfectant, to prevent buildup of tactile and odor cues.

### Visual cue manipulation recording sessions

For the four animals initially habituated to the square environment, on some days, especially when HD cells or grid cells were suspected in the initial baseline recording session, cells were recorded in a subsequent 20 min recording session where a second identical white cardboard sheet was placed along the north wall of the environment (cue B; AB session). This session was followed by a final 20 min recording session where only cue A was present (session order: A1, AB, A2). As with the geometry manipulation sessions, animals were removed from the recording environment between sessions, and the floor was wiped down to prevent buildup of uncontrolled local cues.

### Initial cell classifications with a generalized linear model

Cells were classified as encoding up to four behavioral variables using 10-fold cross-validation with a Poisson generalized linear model (GLM)[10,26]. Those variables were: allocentric HD, egocentric bearing of the environment center, egocentric distance of the environment center, and linear speed. For each model, the firing rate vector $r$ for one cell across the full recording session was modeled as follows:

$$r = \exp\left(\sum_i X_i^T \beta_i\right) \tag{1}$$

where $X$ is a matrix containing animal state vectors for a single behavioral variable over time points $T$, $\beta$ represents the parameter vector for that behavioral variable (akin to a tuning curve), and $i$ indexes across behavioral variables included in the model. The parameter vectors were optimized by maximizing the log-likelihood $l$ of the real spike train $n$ given the estimated rate vector $r$ across time points $t$:

$$l = \sum_t n_t \log(r_t) - r_t - \log(n_t!) \tag{2}$$

A small smoothing penalty, $P$, was added to the objective function to avoid artifacts and overfitting, which penalized differences between adjacent bins of each parameter vector:

$$P = \sum_i S \sum_j \frac{1}{2} * (\beta_{i,j+1} - \beta_{i,j})^2 \tag{3}$$

Here, $S$ is a smoothing hyperparameter (set to 20 for all variables), $i$ indexes over variables, and $j$ indexes over response parameters for a given variable. Response parameters were estimated by minimizing $(P - l)$ using SciPy's *optimize.minimize* function. Thirty bins were used for center bearing and allocentric head direction parameter vectors, and ten bins were used for center distance and linear speed.

Data for a session was split into training (9/10 of the session) and test (1/10 of the session) data ($k = 10$ folds). Parameter vectors were computed by minimizing the objective function on the training data using the full model with all four variables, in order to reduce potential correlative artifacts between independent variables[68]. Log-likelihood values for all possible variable combinations were computed. This procedure was repeated until all parts of the data had been used as test data.

Model selection followed a forward selection procedure[26]. Briefly, the log-likelihood values from the best two-variable model were compared to those from the best one-variable model. If the two-variable model showed significant improvement from the one-variable model (using a one-sided Wilcoxon signed-rank test), then the best three-variable model was compared to the two-variable model, and so on. If the more complex model was not significantly better, the simpler model was chosen. If the chosen model performed significantly better than a model that only included the cell's mean firing rate, the chosen model was used as the cell's classification. Otherwise, the cell was marked 'unclassified'.

### Local vs. global GLM frameworks

To determine the response of EB cells to local vs. global geometry, we created a model using the Poisson GLM that took into consideration the distance and bearing of the two walls closest to the animal at any given point during the recording session (discussed in detail in ref.[28]). We first calculated the egocentric distance and egocentric bearing of the closest point along each of the two closest walls. For each wall, we then created two animal state vectors $X_{bearing_j}$ and $X_{dist_j}$, where $j$ indicates measurements made relative to the $j$th closest wall, that specified the bearing and distance of that wall at every time point in the session. As with the classification GLM, 30 bins for bearing and 10 bins for distance were used. HD and linear speed were also included in the model. We then solved for the optimal parameter vectors $\beta_{bearing}$ and $\beta_{dist}$ (along with HD and speed parameters) by optimizing the GLM as noted above, this time modeling the cell's firing rate as:

$$r = \sum_j (\exp(X_{dist_j}^T \beta_{dist}) * \exp(X_{bearing_j}^T \beta_{bearing})) * \exp\left(\sum_i X_i^T \beta_i\right) \tag{4}$$

such that the cell's response to the bearing of each wall is scaled by its response to the distance of each wall and then summed before being multiplied by the responses to the other variables (HD and speed; indexed by $i$). The centroid model for each cell was created in the same way as the classification GLM, but without the smoothing component, estimating parameters for center-bearing, center-distance, HD, and speed. Because both two-wall and centroid models involved estimating the same number of parameters, we could compare them based on log-likelihood alone without imposing penalties on free parameters.

To test the explanatory power of each model, we trained and tested the models using data from the full session. This procedure involved the computation of a Globality Index (GI), which assesses the

relative model fits for the two-wall and centroid models:

$$GI = \frac{l_{\text{center}} - l_{\text{two-wall}}}{l_{\text{center}} + l_{\text{two-wall}}} \qquad (5)$$

where $l_{\text{center}}$ and $l_{\text{two-wall}}$ represent the log-likelihood increase for each model (in bits/spike) compared to a model that only contains the cell's mean firing rate. GI can theoretically range from -1 (only wall-tuned) to +1 (only center-tuned), although due to the collinearity of centroid and wall variables, they tend to be closer to 0.

### Four-fold symmetry analyses

Coding of the inherent four-fold symmetry of a square environment by EB cells was assessed using the following measures:

1. **Four-fold symmetry of the HD tuning curve**. EB cells sometimes display HD tuning curves with four discrete peaks spaced 90° apart in the square environment. To assess this property, we computed an autocorrelation function for each cell's HD tuning curve. This involved creating a copy of the tuning curve, and correlating it with the original curve at all different possible directional lags (across bins of the tuning curve). The resulting autocorrelation function was used to determine symmetry scores (described below).

2. **Four-fold HD x location correlation structure**. EB cells sometimes display four discrete firing locations in the square environment, each associated with a different HD. To assess the discreteness and four-fold symmetry of these firing fields in the HD domain, we created locational firing rate heatmaps for each EB cell based on time points when the animal was facing a particular HD ($\pm 30°$). Rate maps were created for HDs from 0–360° in 3° increments, after which the rate maps for different HDs were correlated with each other to produce a correlation matrix. A cell with discrete firing fields associated with four equally spaced HDs would be expected to show four discrete 'blocks' of high correlation values along the diagonal of the matrix, with each having a width of approximately 90°. An autocorrelation was computed for the central 90° of this correlation matrix by shifting it along its main diagonal, which was used to calculate symmetry scores (described below).

3. **Four-fold radial symmetry of firing field placement**. EB cells with four-fold symmetry not only have four discrete firing fields associated with four equally spaced HDs, but due to the geometric structure of the square environment, those firing fields are distributed radially about the center of the environment. To assess four-fold symmetry in the radial placement of EB cell firing fields, we created a GLM for each EB cell which attempted to recreate the spike train using a 1-dimensional (1D) distance function and a 1D rotation function (along with allocentric HD). The distance function could be projected across the environment to create a pseudo-2D rate map, which could be rotated about the center of the environment according to the animal's HD (Supplementary Fig. 3). The rotation amount associated with each HD is dictated by the rotation function. A cell without four-fold symmetry would be expected to have a linear rotation function that changes smoothly along with the animal's HD; however, a cell with four-fold radial symmetry would be expected to have a step-wise rotation function that 'snaps' to a new rotation every 90°. These distance and rotation functions were optimized using the same GLM optimization framework used in the classification GLM, with 30 bins used for each variable. An additional penalty was imposed on the mean vector length of the rotation function to encourage sampling of all possible rotations. Four-fold symmetry of the optimized rotation function was assessed by first detrending the function by subtracting a linear range of

angles from 0° to 360°, after which an autocorrelation was computed for the detrended rotation function.

### Symmetry scores

To assess the four-fold symmetry of each 1D autocorrelation function, we took the lowest correlation value at 90°, 180°, or 270° and subtracted the highest correlation value at 45°, 135°, 225°, or 315°. To assess three-fold symmetry, we took the lowest correlation value at 120° or 240° and subtracted the highest correlation value at 60°, 180°, or 300°. Similarly, two-fold symmetry was assessed by subtracting the highest correlation value at 90° or 270° from the correlation value at 180°, while one-fold symmetry was simply computed as the difference between the correlation value at 0° (always 1) and the correlation value at 180°.

To assess significance of tuning to each degree of symmetry, we compared the symmetry scores of real EB cells to scores computed from EB cell spike trains that had been randomly shifted relative to the behavioral data (shuffle distribution). Spike trains were randomly shifted by at least 30 s, with spikes that extended beyond the end of the session wrapped to the beginning. This procedure was repeated 100 times for each EB cell, and the 95th percentile for each symmetry score was used as a cutoff for classifying EB cells as having significant symmetrical tuning.

### Assessment of trough vs. peak HD tuning

POR HD cells have been previously found to fall into one of two groups: peak cells, which have their firing rate maximum in the general direction of the familiar cue card; and trough cells, which have their firing rate minimum in the direction of the cue card[23]. These cell types can be distinguished by fitting both an upright (sharp peak) and inverted (sharp trough) von Mises function to each cell's tuning curve and calculating the difference in $R^2$ fit ($R^2_{\text{upright}} - R^2_{\text{inverted}}$), with peak cell tuning curves better fit by an upright von Mises function and trough cell tuning curves better fit by an inverted von Mises function. To test if RSC or MEC/PaS HD cells showed the same pattern, we compared upright vs. inverted von Mises $R^2$ values for HD cells with maximal firing directions toward the cue card (180° <maximal direction <360°) or away from the cue card (0° <maximal direction <180°).

### Assessment of HD cell bidirectionality

To assess if HD cells fired in two opposite directions in the AB experiment, we computed a bidirectionality index (discussed in detail in[23]). Briefly, two tuning curves were constructed for each HD cell: one based on the animal's actual HD; and one where the animal's HD had been doubled first. Symmetrical bidirectional distributions can be transformed into unidirectional distribution by doubling the associated angles. The bidirectionality index was then computed as follows:

$$\text{Bidirectionality Index} = (\text{MVL}_{\text{doubled}} - \text{MVL}_{\text{normal}})/(\text{MVL}_{\text{doubled}} + \text{MVL}_{\text{normal}})$$

$$(6)$$

### Cue modulation measures

To determine the extent to which HD cells incorporated the second cue card in the AB session, we fit a bidirectional von Mises function (two peaks or troughs separated by 180°) to each cell's HD tuning curve from the AB session[23]. Trough fits were used only for POR HD cells with maximal firing directions oriented away from the cue card, and RSC and MEC/PaS cells were fit with upright von Mises functions as they did not exhibit trough tuning. Modulation by cue A was calculated by finding the von Mises peak or trough that was closest to the cell's A1 peak or trough, then computing the firing rate difference between that peak or trough and the minimum or maximum of the fit curve, respectively. This firing rate difference was transformed into a

modulation index (MI) by dividing it by the maximum firing rate of the fit curve (fr = firing rate):

$$MIA = (peak\_fr_A - min\_fr(fit\_curve))/max\_fr(fit\_curve) \quad (7)$$
$$[for\ peak\ or\ non-POR\ cells]$$

OR

$$MIA = (max\_fr(fit\_curve) - trough\_fr_A)/max\_fr(fit\_curve) \quad (8)$$
$$[for\ POR\ trough\ cells]$$

where A indicates the portion of the tuning curve associated with cue A. The MI for cue B was calculated by performing the same computation on the peak or trough 180° opposite.

### Assessment of waveform width
As discussed above, waveforms were captured at 32 kHz for 1 ms, such that each waveform had 32 samples. For each cell, the average waveform for each channel of the associated tetrode was computed, and the channel with the largest amplitude was chosen for analysis. The locations of the peak and trough were estimated using a cubic spline (upsampled 100x) to give a more precise estimate of waveform width, which was then calculated as the temporal difference between the peak and trough.

### Egocentric tuning curves and classification
Egocentric bearing tuning curves were constructed using 12° bins. For each cell, a center-bearing tuning curve was constructed by dividing the number of spikes associated with each bin by the amount of time that bin was occupied. The mean vector length (MVL) and mean angle of the tuning curve were computed to establish the cell's tuning strength and preferred bearing, respectively. A cell was considered an egocentric bearing cell if it: i) passed the GLM classification procedure for center-bearing tuning (discussed above); ii) had an MVL that passed the 99th percentile of a within-cell shuffle distribution (discussed below) or 0.10, whichever was larger; and iii) had a peak firing rate in its center-bearing tuning curve that exceeded 1 Hz.

### Allocentric HD tuning curves and classification
Allocentric HD tuning curves and classifications followed largely the same procedure as egocentric bearing tuning outlined above, but using the animal's allocentric HD. However, to prevent RSC cells with strong four-fold symmetry being incorrectly classified as unidirectional "HD cells," two additional criteria were imposed to emphasize cells with unidirectional firing preferences: i) a strict MVL minimum threshold of 0.15 was used in addition to the 99th percentile shuffle cutoff; and, ii) HD cells were only included in the AB experiment if their preferred firing direction changed by <45° between the A1 and A2 sessions.

### Allocentric location firing rate maps
The animal's two-dimensional location throughout the recording session was divided into 4 cm × 4 cm bins. For each cell, the number of spikes associated with each bin was divided by the amount of time the bin was occupied. The resulting firing rate map was smoothed with a Gaussian filter.

### Assessment of firing along inserted walls in L-shaped environment
To assess the difference in firing rates between square and L-shaped environments, firing rate maps for each session were first range normalized to the highest and lowest value across both square and L-shape sessions. Because some EB cells have their highest firing rates in the center of the environment and their lowest firing rates along the boundaries[10,21], the firing rate maps for cells with this pattern were inverted by subtracting the firing rate in each bin from the overall maximum firing rate[28]. For all cells, the difference between the square and L-shape rate maps in each spatial bin was then computed. The mean change within 20 cm of the inserted walls was used to assess increased firing in the vicinity of the inserted walls, while the mean change > 20 cm away was used as a control measure.

### Grid cell classifications
After computing a cell's allocentric firing rate map, it was used to compute a grid score[12]. A 2-dimensional autocorrelation was computed by correlating a copy of the rate map with the original rate map at all possible spatial shifts. A cell with hexagonally periodic firing fields would be expected to show a ring around the center of the autocorrelation with six evenly spaced peaks. For each cell, we identified the most probable inner and outer radii of this ring, and then computed a 1-dimensional autocorrelation by correlating a copy of the ring with the original ring at 3° rotational offsets from 0°to 180°. A grid score was then computed from the 1D autocorrelation by taking the lowest correlation value at 60° or 120° and subtracting the highest correlation value at 30°, 90°, or 150°. Cells with grid scores > 0.4 were considered grid cells.

### Shuffling procedure for cell classifications
Each cell's spike train was randomly shifted by at least 30 s, with time points beyond the end of the session wrapped to the beginning, to offset the spike data from the behavioral data while maintaining its temporal structure. Relevant tuning scores were then computed based on the shifted spike train. This procedure was repeated 400 times for each cell, and a within-cell 99th percentile cutoff was used to determine tuning significance for individual cells.

### Statistics
Statistical analyses were performed using Python code. All tests were nonparametric and two-sided (except for GLM classification comparisons which were one-sided[10,26]) and used an α level of 0.05. Paired comparisons were made using Wilcoxon signed-rank tests, while unpaired comparisons used a rank-sum test.

### Reporting summary
Further information on research design is available in the Nature Portfolio Reporting Summary linked to this article.

## Data availability
Data needed to reproduce the main figure plots, as well as example cell data for running symmetry analyses that are central to this paper, are available at github.com/hasselmonians/LaChance_Hasselmo_POR_RSC (doi.org/10.5281/zenodo.13502206). Data needed to reproduce the main results and figure panels are provided with the manuscript as Source Data. Source data are provided with this paper.

## Code availability
All code needed to reproduce the main figure plots, as well as custom code for running symmetry analyses central to the conclusions of this paper, is available at github.com/hasselmonians/LaChance_Hasselmo_POR_RSC (https://doi.org/10.5281/zenodo.13502206). Code for GLM classification and local vs. global analyses have been uploaded previously at github.com/taube-lab/POR_GLM (https://doi.org/10.5281/zenodo.3173242) and github.com/taube-lab/LaChance_Taube_Curr_Biol_2023, respectively.

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

## Acknowledgements

This work was supported by the National Institutes of Health grant number R01 MH120073 (M.E.H.), and by the Office of Naval Research MURI N00014-19-1-2571 (M.E.H.).

## Author contributions

P.A.L. and M.E.H. conceptualized and designed the experiments. P.A.L. performed the experiments, wrote and ran the analyses, and wrote the initial manuscript draft. P.A.L. and M.E.H. revised the manuscript.

## Competing interests

All the authors declare no competing interests.
