## [Peer Review File · Nature Communications]

REVIEWER COMMENTS

Reviewer #1 (Remarks to the Author):

The manuscript by LaChance and Hasselmo presents a follow-up study to their previously published works in POR (LaChane et al., 2019, 2022, and LaChance and Taube, 2023), extending the same framework to the RSC. By comparing single neuron activity between the POR and RSC side by side, this manuscript aims to address the important question of which brain region encodes local geometric features, versus the concept of global geometry of the environment in an egocentric reference frame. The authors first record neural activity from the animals in a square environment and found that the tuning of RSC EB cells showed a much stronger 4-fold symmetry than the POR EB cells. Further recording in an L-shaped environment confirmed the local vs. global encoding of geometric features in RSC and POR EB cells, respectively. Finally, the authors compared the encoding of head direction and found that although both POR and RSC HD cells showed bidirectional tuning upon adding an additional visual cue, narrow-waveform HD cells in the RSC remained unidirectional. Although the experiments and analyses are robust and well considered, I have some concerns regarding whether the conclusion is supported by the current experiments.

1. My primary concern is whether the animal is indeed using the centroid to form an estimation of the global environmental geometry. Although the centroid is mathematically equivalent to the sum of the vectors from boundaries, it is still unclear to me whether the effects reported in the L-shaped environment, both in this study and their previous study, are only specific to this environment. In the authors' previous *Current Biology* paper, they could not recapitulate the same finding of POR EB neurons in a rectangle environment. They did identify some confounding factors that may have contributed to the observations in the L-shaped environment, but did not sufficiently address these in the current study. I suggest the authors perform additional experiments in at least one other differently shaped arena, such as an asymmetric triangle or trapezoid, where the bearings to the wall and the centroid are sufficiently de-correlated, to see whether this finding can be repeated across geometrically distinct environments.

2. Related to the use of the centroid, in a more extreme case, one can further morph the L-shaped environment into a V-shaped environment, in which the centroid could lie outside of the environment. Could the authors provide discussion regarding how POR neurons might adapt their tuning and estimate the global geometry when the centroid is outside of the arena?

3. Figures 1 and 2: The authors conclude that POR EB cells show circular symmetry in their tuning in the square environment, implying that the directional signal of POR EB cells forms a continuous and smooth distribution around the centroid. However, this is not the case for individual neurons.

For example, in Figure 2A, it appears that a significant portion of POR EB cells (the bottom half) exhibit a unimodal or bimodal distribution regarding their head direction tuning. It seems that POR EB cells comprise a mixture of 1-fold, 2-fold, 3-fold, and 4-fold neurons, and the circular symmetry could result from averaging across these neurons.

4. In the RSC, although the top 50% of EB cells (Figure 2A) exhibit very strong 4-fold symmetry, the bottom half of the RSC EB cells resemble more of a 'circular symmetry'. Perhaps there are two groups of RSC EB cells, with one encoding local features and the other encoding global geometry? Given that there were over 200 EB neurons in the RSC, the authors could potentially split them into halves and explore this possibility.

5. Definitions of neurons: The authors define EB cells and HD cells using a GLM framework containing egocentric (center) bearing, egocentric distance, allocentric head direction, and linear speed. For both EB and HD cells used in the study, it would be helpful for the authors to show the full spectrum of these neurons' conjunctive coding properties. For example, what proportion of EB cells showed mixed selectivity for both egocentric bearing and head direction, and what proportion of EB cells showed mixed selectivity for all four behavioral variables, etc.

Minor

1. Regarding the definition of EB cells: As there is no allocentric position in the GLM, I was curious whether there were any EB cells that showed significant place coding? I am referring to cells that fire in one or two particular locations, not egocentric boundary cells (which likely have significant spatial information). If so, what's the percentage of such cells in POR and RSC?

2. Figures 1-2: When possible, it would be helpful to add a shuffle line to the existing plots to facilitate the comparison between POR and RSC, as well as to the random condition.

3. Line 118: one-to-one relationship?

4. ED Figure 12: it would be good to draw a line at 0.2 ms to make the reader easier to see how narrow vs. wide waveform neurons were defined. Same as Figure 5D.

Reviewer #2 (Remarks to the Author):

LaChance and Hasselmo compare the encoding of egocentric variables, such as bearing relative to the walls and corners of a room, to allocentric variable such as global position within the room, or absolute heading, across postrhinal (POR), and retrosplenial (RSC) cortex. They also provide some reference data from entorhinal cortex. Together, their results provide new insights into the encoding of these different types of sensory and spatial information across brain regions underlying navigation.

The experiments and analyses appear done at a state-of-the art level, and results are of great interest in the field.

My main comment on this manuscript is that a clearer presentation of the findings could make the results more accessible and would likely lead to more scientists incorporating these results into their thinking. A single figure contrasting the major findings for POR, RSC and ERC side-by-side, either re-using panels from the paper, or in a cartoon format would for instance spare others from having to assemble such a figure when they wish to refer to this paper in their own lab meetings etc.

Minor comments:

-Fig 1 A: The polarizing cue is not labelled or referred to anywhere. It also is not mentioned in the intro, at least not near line 52 where one would expect it.

-Fig.1I&J: it is not immediately clear how this correlation matrix was computed. Referring to the specific methods paragraph would help, and if possible, see if a short description in the caption or main text is possible.

- Line 57: Since this sentence talks not just about the present study, it would be clearer to say '(and therefore, for square environments, encode ...'.

-There are ongoing debates in the field about whether there are functionally defined discrete cell types, or whether coding properties lie on a continuum. If the N in this study allows, it a supplementary analysis that simply plots a histogram or scatter plot of how distinctly all recorded cells code for the studied ego/allo variable (e.g. via entropy or some other measure of coding accuracy) would be useful.

We thank the Reviewers for their thorough reading of our manuscript and their overall positive assessment of the experiments and analyses. We have made many changes to the manuscript in response to the Reviewer's critiques which we believe have significantly improved and strengthened the paper. In the following text, Reviewer comments are italicized while our responses are in normal font.

Reviewer #1 (Remarks to the Author):

The manuscript by LaChance and Hasselmo presents a follow-up study to their previously published works in POR (LaChane et al., 2019, 2022, and LaChance and Taube, 2023), extending the same framework to the RSC. By comparing single neuron activity between the POR and RSC side by side, this manuscript aims to address the important question of which brain region encodes local geometric features, versus the concept of global geometry of the environment in an egocentric reference frame. The authors first record neural activity from the animals in a square environment and found that the tuning of RSC EB cells showed a much stronger 4-fold symmetry than the POR EB cells. Further recording in an L-shaped environment confirmed the local vs. global encoding of geometric features in RSC and POR EB cells, respectively. Finally, the authors compared the encoding of head direction and found that although both POR and RSC HD cells showed bidirectional tuning upon adding an additional visual cue, narrow-waveform HD cells in the RSC remained unidirectional. Although the experiments and analyses are robust and well considered, I have some concerns regarding whether the conclusion is supported by the current experiments.

We thank the Reviewer for their assessment of these experiments and analyses as “robust and well considered,” and hope our responses to the following points are sufficient to address the Reviewer's concerns regarding the experimental conclusions.

1. My primary concern is whether the animal is indeed using the centroid to form an estimation of the global environmental geometry. Although the centroid is mathematically equivalent to the sum of the vectors from boundaries, it is still unclear to me whether the effects reported in the L-shaped environment, both in this study and their previous study, are only specific to this environment. In the authors' previous Current Biology paper, they could not recapitulate the same finding of POR EB neurons in a rectangle environment. They did identify some confounding factors that may have contributed to the observations in the L-shaped environment, but did not sufficiently address these in the current study. I suggest the authors perform additional experiments in at least one other differently shaped arena, such as an asymmetric triangle or trapezoid, where the bearings to the wall and the centroid are sufficiently decorrelated, to see whether this finding can be repeated across geometrically distinct environments.

The Reviewer's point is well taken, and we agree that further experiments (i.e., asymmetric triangle or trapezoidal environments) will be important and necessary to fully characterize the exact features of the environment driving spatial firing among POR EB cells. Despite our agreement with the Reviewer's point, we believe that the question of whether POR cells are truly representing the centroid as a geometric concept is beyond the scope of the current manuscript (and is not a claim we are trying to make here), which is intended to compare the responses of

RSC and POR cells to local geometric features that may give rise to neural coding of rotational symmetry. Use of the centroid in our analyses (which the Reviewer acknowledges is equivalent to the sum of vectors from boundaries) is simply a convenient way to capture the firing properties of EB cells that have diffuse responses to environmental boundaries at a range of distances (i.e., global geometric coding), which appear to differ qualitatively from EB cells that respond to particular boundary features when they occupy a very specific distance and direction. In an attempt to clarify this point, we have removed most mentions of ‘center-bearing’ and ‘centroid’ from the manuscript (though a few mentions are retained that refer to previously published results or existing centroid-based analyses), instead focusing on the distinction between global and local coding of geometry, and acknowledging the possibility that global encoding could take multiple forms. This includes changing the axis label ‘center-bearing’ to ‘egocentric bearing’ in relevant plots, and adding the following passage to the Discussion (also related to point 2 below; **line 233**):

“However, while the current study reinforces the notion that that POR EB cells are sensitive to a more global account of environmental geometry than RSC EB cells, future experiments in more complex environments (e.g., trapezoid or asymmetric triangle) will be necessary to determine if POR EB cells are truly computing the environment centroid or if they more closely follow some other boundary-encoding framework. For example, in a V-shaped or similar environment where the centroid lies outside the navigable space, POR EB cells may encode the medial axis of the shape (Cheng and Gallistel, 2005) instead of representing a single centroid, or could potentially display more local geometric coding than that observed in square and L-shaped environments. It has been previously demonstrated that POR EB cells can vary in their tuning to local vs. global aspects of environmental geometry (LaChance and Taube, 2023), so different cells may exhibit distinct responses.”

Other places where references to center/centroid have been removed or revised:

Abstract

“...while POR cells encode a more global account of boundary geometry (environment centroid).”

CHANGED TO

““...while POR cells encode a more global account of boundary geometry.”

Line 69

“...have suggested that POR EB cells are sensitive to the global geometry of the surrounding environment (and therefore encode the environment center)...”

CHANGED TO

“...have suggested that POR EB cells are sensitive to the global geometry of the surrounding environment (equivalent to coding of the environment center)...”

Line 73

“...and contrast this with global coding of the environment in the form of a response to the centroid of the environment.”

CHANGED TO

“...and contrast this with global coding of the environment’s extended boundary geometry.”

Line 110

“...while POR EB cells exhibit circular symmetry...”

CHANGED TO

“...while POR EB cells tend to lack periodic symmetry...”

2. Related to the use of the centroid, in a more extreme case, one can further morph the L-shaped environment into a V-shaped environment, in which the centroid could lie outside of the environment. Could the authors provide discussion regarding how POR neurons might adapt their tuning and estimate the global geometry when the centroid is outside of the arena?

We agree with the Reviewer that it will be important to record in environments where the centroid falls outside the environment, in order to fully characterize the geometric tuning properties of POR EB cells. We have added text to the discussion in order to address this point and to suggest possibilities for how POR EB cells may encode geometry in such an environment (see response the paragraph added to the Discussion in the response to point 1 above).

3. Figures 1 and 2: The authors conclude that POR EB cells show circular symmetry in their tuning in the square environment, implying that the directional signal of POR EB cells forms a continuous and smooth distribution around the centroid. However, this is not the case for individual neurons. For example, in Figure 2A, it appears that a significant portion of POR EB cells (the bottom half) exhibit a unimodal or bimodal distribution regarding their head direction tuning. It seems that POR EB cells comprise a mixture of 1-fold, 2-fold, 3-fold, and 4-fold neurons, and the circular symmetry could result from averaging across these neurons.

We thank the Reviewer for this important point. To address it, we expanded our symmetry analyses to include assessments of 1-fold, 2-fold, 3-fold, and 4-fold symmetry. We also created a shuffle distribution of symmetry scores based on shuffled spike trains (100 per cell; related to minor point 2 below) and defined a 95th percentile cutoff to classify individual cells as showing significant tuning to each symmetry type. These additional symmetry scores and shuffle cutoffs are now included in Figure 2.

The percent of cells that passed 95th percentile shuffle distribution was, for POR: 1-fold: 26%, 2-fold: 8%, 3-fold: 2%, and 4-fold: 7%; for RSC: 1-fold: 4%, 2-fold: 7%, 3-fold: <1%, 4-fold: 52%. Overall, the only proportions that were substantially greater than the expected 5% were 1-fold symmetry for POR and 4-fold symmetry for RSC. The significant proportion of 1-fold symmetry cells in POR is indicative of tuning properties that deviate from perfect circular symmetry but not in a multi-fold periodic way (e.g., an EB cell with a broad preference for a particular HD, or a cell with a bumpy but non-periodic tuning curve). We do not believe that this result conflicts with our conclusion that POR EB firing overall reflects circular symmetry, and instead it demonstrates that, when POR cells do deviate from circular symmetry, they do so in a way that avoids multi-fold periodic firing (and therefore are unlikely to be responding to symmetrical geometric features). In contrast, when RSC cells deviate from circular symmetry, they tend to do so in a way that captures the rotational symmetry of local geometric features (i.e., four-fold symmetry). We have added the following text to the manuscript to expand on these points:

Line 89

“In addition to the expected four-fold symmetry, we also computed scores for one-fold, two-fold, and three-fold symmetry (Fig. 1E). While we expected RSC EB cells to display strong four-fold symmetry related to local geometric features, we expected POR EB cells to lack strong periodic symmetry, which may manifest in a larger population of cells showing one-fold symmetry (i.e., any deviation from circular symmetry would be non-periodic).”

Line 99

“For RSC EB cells, 109/210 (52%) of RSC EB cells crossed this threshold for four-fold symmetry, while only 6/85 (7%) of POR EB cells did (Fig. 2G). In contrast, while only 8/210 (4%) of RSC EB cells showed significant one-fold symmetry, 22/85 (26%) of POR cells did (Fig. 2G). Neither region exhibited strong evidence for two-fold (POR: 8%, RSC: 7%) or three-fold (POR: 2%, RSC: <1%) symmetry. ... Overall, RSC EB cells tend to exhibit strong four-fold radial symmetry in a square environment, indicative of coding for local geometric features such as walls and corners, while POR EB cells tend to lack periodic symmetry, indicative of coding for global properties of environmental geometry.”

4. In the RSC, although the top 50% of EB cells (Figure 2A) exhibit very strong 4-fold symmetry, the bottom half of the RSC EB cells resemble more of a 'circular symmetry'. Perhaps there are two groups of RSC EB cells, with one encoding local features and the other encoding global geometry? Given that there were over 200 EB neurons in the RSC, the authors could potentially split them into halves and explore this possibility.

The Reviewer is correct that the bottom half of RSC EB cells do not display significant four-fold symmetry when compared to a shuffle distribution and may reflect more of a ‘circular symmetry’. However, comparing the four-fold symmetry scores of these ‘non-symmetric’ RSC EB cells with those of the POR EB cells still reveals that the ‘non-symmetric’ RSC EB cells tend to exhibit significantly more four-fold symmetry than the population of POR EB cells (i.e., are more ‘rightward shifted’ in the new panel 2H, which show histograms of 4-fold symmetry scores for POR and RSC EB cells that had symmetry scores < 0). In addition, there do not seem to be two separate populations of RSC EB cells in terms of four-fold symmetry, but rather the cells vary along a continuum. Therefore, we believe the lack of significant symmetry scores for these RSC cells may simply reflect a lack of power in the statistical analyses for detecting subtle responses to four-fold symmetry. While we are hesitant to ascribe a label of ‘circularly symmetrical’ to RSC EB cells that do not meet four-fold symmetry criteria, we have added the following text to the manuscript to address this point:

Line 104

“While 48% of RSC EB cells did not pass the threshold for strong four-fold symmetry, those non-symmetrical RSC cells still had significantly higher four-fold symmetry scores than the full POR EB cell population ($Z = 5.41$, $P = 6.14e-8$; Fig. 2H), suggesting that even RSC cells with subthreshold four-fold symmetry scores are distinct from POR cells in their encoding of environmental symmetry.”

5. Definitions of neurons: The authors define EB cells and HD cells using a GLM framework containing egocentric (center) bearing, egocentric distance, allocentric head direction, and linear speed. For both EB and HD cells used in the study, it would be helpful for the authors to show the full spectrum of these neurons' conjunctive coding properties. For example, what proportion of EB cells showed mixed selectivity for both egocentric bearing and head direction, and what proportion of EB cells showed mixed selectivity for all four behavioral variables, etc.

This is a good point, and we have added additional supplemental panels (Extended Data Figure 1A-B) which shows the full spectrum of combinations of multiple variable classifications for individual neurons in POR and RSC. As has been demonstrated previously, POR and RSC cells tend to be highly conjunctive and often respond to multiple behavioral variables simultaneously. We now specifically address this for EB cells in the manuscript text:

Line 58

“Many EB cells in both regions were classified as encoding at least one other behavioral variable, including HD (POR: 60%, RSC: 33%), egocentric distance of the environment center/boundaries (POR: 22%, RSC: 21%), and linear speed (POR: 24%, RSC: 24%) or combinations of these variables as shown in Extended Data Fig. 1A-B.”

Minor

1. Regarding the definition of EB cells: As there is no allocentric position in the GLM, I was curious whether there were any EB cells that showed significant place coding? I am referring to

cells that fire in one or two particular locations, not egocentric boundary cells (which likely have significant spatial information). If so, what's the percentage of such cells in POR and RSC?

The Reviewer is correct that egocentric boundary cells can have high spatial information content, which can make it difficult to assess traditionally defined place coding. To attempt to find cells with particularly high degrees of spatial information content, we created firing rate maps for EB cells that passed a 99th percentile within-cell shuffle cutoff for spatial information content. These included 5/85 POR EB cells (6%) and 29/210 RSC EB cells (14%). However, the rate maps are generally unlike those of traditional place cells, with firing fields that tend to follow the full boundary of the environment (or that appear to be inhibited by the full boundary of the environment in some cases). Cases where cells fire in a more restricted portion of the environment tend to have conjunctive EB x HD tuning, which biases firing toward one side of the environment, so it is unlikely that any of these cells exhibit true 'place coding.' Because the question of explicit place coding is somewhat beyond the scope of the current set of experiments (despite being quite interesting!), we have not made any related changes to the manuscript, but are including the rate maps here for the Reviewer's convenience:

POR EB cells with high spatial information

RSC EB cells with high spatial information

2. Figures 1-2: When possible, it would be helpful to add a shuffle line to the existing plots to facilitate the comparison between POR and RSC, as well as to the random condition.

We thank the Reviewer for this important point, and have now added a 95th percentile shuffle line to the symmetry score plots in order to properly demonstrate which cells show significant symmetrical coding.

3. Line 118: one-to-one relationship?

We have edited this line in the following way:

“...suggesting against a one-to-one relationship between EB cell and grid cell encoding of environmental geometry.”

CHANGED TO

“...suggesting distinct mechanisms underlying EB cell and grid cell encoding of environmental geometry.”

4. ED Figure 12: it would be good to draw a line at 0.2 ms to make the reader easier to see how narrow vs. wide waveform neurons were defined. Same as Figure 5D.

We have now added a line at 0.2 ms (in both ED Figure 12 and Figure 5D) to indicate how narrow vs. wide waveforms were defined.

Reviewer #2 (Remarks to the Author):

LaChance and Hasselmo compare the encoding of egocentric variables, such as bearing relative to the walls and corners of a room, to allocentric variable such as global position within the room, or absolute heading, across postrhinal (POR), and retrosplenial (RSC) cortex. They also provide some reference data from entorhinal cortex. Together, their results provide new insights into the encoding of these different types of sensory and spatial information across brain regions underlying navigation.

The experiments and analyses appear done at a state-of-the art level, and results are of great interest in the field.

We thank the Reviewer for their assessment of our experiments and analyses as ‘state-of-the-art’ and our results as ‘of great interest in the field.’ We hope the following changes made to the manuscript will be satisfactory in alleviating the Reviewer’s concerns.

My main comment on this manuscript is that a clearer presentation of the findings could make the results more accessible and would likely lead to more scientists incorporating these results into their thinking. A single figure contrasting the major findings for POR, RSC and ERC side-by-side, either re-using panels from the paper, or in a cartoon format would for instance spare others from having to assemble such a figure when they wish to refer to this paper in their own lab meetings etc.

We thank the Reviewer for this point, and have created a new figure (Figure 6 – Summary of Results) which combines panels from the previous figures in order to provide a succinct summary of results across the different experiments. It also provides an additional schematic of the three symmetry domains we used to assess symmetrical coding. We hope this figure has satisfied the Reviewer's concern.

Minor comments:

-Fig 1 A: The polarizing cue is not labelled or referred to anywhere. It also is not mentioned in the intro, at least not near line 52 where one would expect it.

We have now labeled the polarizing cue in Figure 1A, and have added the following near line 52: "...with a single polarizing cue (large white card) placed along the south wall."

-Fig. II&J: it is not immediately clear how this correlation matrix was computed. Referring to the specific methods paragraph would help, and if possible, see if a short description in the caption or main text is possible.

We have now added a reference to the specific methods paragraph (*Four fold symmetry analyses*), and have added a short description in the main text (new text underlined; **line 76**):

"2) four-fold symmetry in the correlation structure of location preferences across different head directions (computed by creating an array of firing rate maps for periods when the animal faced different HDs and calculating pairwise correlations between the rate maps..."

- Line 57: Since this sentence talks not just about the present study, it would be clearer to say '(and therefore, for square environments, encode ...)'.

As suggested, we have now added the phrase "for square environments" after "and therefore,"

-There are ongoing debates in the field about whether there are functionally defined discrete cell types, or whether coding properties lie on a continuum. If the N in this study allows, it a supplementary analysis that simply plots a histogram or scatter plot of how distinctly all recorded cells code for the studied ego/allo variable (e.g. via entropy or some other measure of coding accuracy) would be useful.

We thank the Reviewer for this important point, as it is of critical importance whether neurons belong to discrete ‘cell types’ or whether they simply have response properties that vary along a continuum. We have added a new Extended Data Figure 2 which provides histograms of various measures of tuning strength for all cells in POR and RSC. These include information content measures derived from the GLM, which attempt to characterize the amount of information each spike communicates about a given variable after taking all four spatial variables into account (ego bearing, ego distance, HD, and speed) and therefore provides a conservative estimate for each cell’s information content related to each variable. We find that all tuning measures tend to vary along a continuum, and that overall individual ‘cell types’ don’t appear to form discrete groups. It is, however, important to keep in mind that cells are classified based on multiple measures as well as cross-validation with the GLM. We have added the following text to the manuscript:

Line 62

“Strength of tuning to each of these variables varied along a continuum from non-tuned to strongly tuned, suggesting that significantly tuned cells may not constitute distinct ‘cell types’ despite showing consistent responses to a given variable (Extended Data Fig. 2).”

REVIEWERS' COMMENTS

Reviewer #1 (Remarks to the Author):

The authors have thoroughly addressed my comments and the paper will add a significant contribution to our understanding of how the brain encodes the external environment.

Reviewer #2 (Remarks to the Author):

The additional clarifications and figures address all of my concerns